# Airflow and optic flow mediate antennal positioning in flying honeybees

**Taruni Roy Khurana, Sanjay P Sane\***

National Centre for Biological Sciences, Tata Institute of Fundamental Research, Bangalore, India

**Abstract** To maintain their speeds during navigation, insects rely on feedback from their visual and mechanosensory modalities. Although optic flow plays an essential role in speed determination, it is less reliable under conditions of low light or sparse landmarks. Under such conditions, insects rely on feedback from antennal mechanosensors but it is not clear how these inputs combine to elicit flight-related antennal behaviours. We here show that antennal movements of the honeybee, *Apis mellifera,* are governed by combined visual and antennal mechanosensory inputs. Frontal airflow, as experienced during forward flight, causes antennae to actively move forward as a sigmoidal function of absolute airspeed values. However, corresponding front-to-back optic flow causes antennae to move backward, as a linear function of relative optic flow, opposite the airspeed response. When combined, these inputs maintain antennal position in a state of dynamic equilibrium.

**\*For correspondence:** sane@ncbs.res.in

**Competing interests:** The authors declare that no competing interests exist.

## Introduction

When flying in unpredictable conditions, sensory cues from a single modality are often unreliable measures of the ambient environmental parameters. For instance, purely optic flow-based measurements of self-motion can be misleading for insects which experience sideslip while flying in a cross-wind. Moreover, reliance on optic flow may be problematic under dimly lit or overcast conditions, or when flying over lakes or deserts which present sparse visual feedback. In such situations, sampling from multiple sensory cues reduces the ambiguity arising from variability in feedback from single modalities (*Wehner, 2003*; *Sherman and Dickinson, 2004*; *Wasserman et al., 2015*). Hence, the integration of multimodal sensory cues is essential for most natural locomotory behaviours, including insect flight manoeuvres (*Willis and Arbas, 1991*; *Frye et al., 2003*; *Verspui and Gray, 2009*).

For flight control, the importance of optic flow cues detected by compound eyes is well-documented in diverse insects, including honeybees (*Srinivasan et al., 1996*; *Baird et al., 2005*), bumblebees (*Baird et al., 2010*; *Dyhr and Higgins, 2010*), *Megalopta* (*Baird et al., 2011*) and *Drosophila* (*David, 1982*; *Duistermars et al., 2009*). In recent years, mechanosensory feedback from antennae has also emerged as a key sensory input for insect flight (*Sane et al., 2007*; *Yorozu et al., 2009*; *Krishnan et al., 2012*; *Fuller et al., 2014*). This feedback is transduced primarily by two sets of mechanosensors - the chordotonal *Johnston's organ* (JO) (*Gewecke, 1974*) which senses a wide range of stimuli from high-frequency antennal vibrations to low-frequency ambient airflow or gravity (*Yorozu et al., 2009*; *Dieudonné et al., 2014*), and the antennal hair plates (or *Böhm's bristles*) which are involved in the reflexive positioning of antennae during flight (*Krishnan et al., 2012*).

The characteristic positioning of the antennae at the onset of flight is ubiquitous in most, if not all, flying insects underscoring the evolutionary significance of its function (*Dorsett, 1962*). Disruption of antennal positioning due to Böhm's bristle ablation or reduction of JO inputs severely impairs flight (*Willis et al., 1995*; *Hinterwirth and Daniel, 2010*; *Sane et al., 2007*). The control of antennal position is thought to be essential for the unambiguous sensing of inputs by the JO

**eLife digest** Insects combine information from different senses to help them navigate during flight. Flying insects see moving images, which the brain can use to measure their speeds. Insect antennae also help to judge speed, as they signal to the brain about the physical forces that result from the insect moving through the air. To accurately detect these forces, and also to detect odors from the surrounding environment, insects must precisely position their antennae as they fly.

To investigate how honeybees use different types of sensory information to position their antennae during flight, Roy Khurana and Sane first placed freely-flying and tethered bees in a wind tunnel. Flying forward causes air to flow from the front to the back of the bee. The experiments revealed that a bee brings its antennae forward and holds them in a specific position that depends on the rate of airflow. As the bee flies forward more quickly (or airflow increases), the antennae are positioned further forward.

Roy Khurana and Sane then investigated how the movement of images across the insect's eyes causes their antennae to change position. This unexpectedly revealed that moving images across the eye from front to back, which simulates what bees see when flying forward, causes the bees to move their antennae backward. However, exposing the bees to both the frontal airflow and front-to-back image motion as normally experienced during forward flight caused the bees to maintain their antennae in a fixed position. This behaviour results from the opposing responses of the antennae to the two stimuli.

Future challenges will be to determine how the brain of a honeybee combines the information from different senses to position the antennae, and to discover what this behaviour implies for insect flight in general.

(*Hinterwirth et al., 2012*). However, the mechanisms underlying this behaviour are not well-understood. Here, we show that the airflow cues sensed by Johnston's organs and the optic flow cues sensed by eyes combine to maintain and control antennal position during flight in the honeybee, *Apis mellifera*. Each input influences antennal position in an opposite manner; frontal airflow causes antennae to move actively forward against the aerodynamic drag, whereas front-to-back optic flow causes them to move backward. The antennal positioning response thus offers a critical readout for understanding how honeybees integrate information about their own motion from airflow and optic flow cues.

## Results

### Flying honeybees bring their antennae forward in response to increasing airspeed

To characterize antennal response to ambient airflow, we provided flying bees with frontal airflow cues as would be experienced by them during forward flight. We then calculated the inter-antennal angle (IAA), defined as the angle between the lines joining the base and tip of each antenna (*Sane et al., 2007*), as a measure of antennal position (*Figure 1A*). When the antennae move backwards, the IAA increases and when the antennae move forward, the IAA decreases. Of the three conventional variables used in such experiments (*David, 1982*), we experimentally set the *windspeed* (*i.e.* velocity of ambient airflow relative to ground) in a calibrated, laminar wind tunnel, whereas the bees controlled their own *airspeed* (*i.e.* velocity of body relative to ambient air) and therefore also their *groundspeed* (*i.e.* velocity of the body relative to ground), which is the vector sum of *airspeed* and *windspeed* (*Figure 1B*). As tethered bees are stationary relative to the ground, their *airspeed* equals *windspeed*. However, when freely flying in air currents, *airspeed* and *windspeed* are independent of each other. Analogous to swimmers swimming within water currents, insects control their *airspeed* relative to the air pocket drifting at some *windspeed*.

Honeybees modulated their antennal position as a function of their absolute airspeed. When presented with fixed airflow but no optic flow, both tethered and freely-flying bees held their antennae at constant IAA throughout a flight bout (e.g. *Figure 1—figure supplement 1A*). As frontal airflow

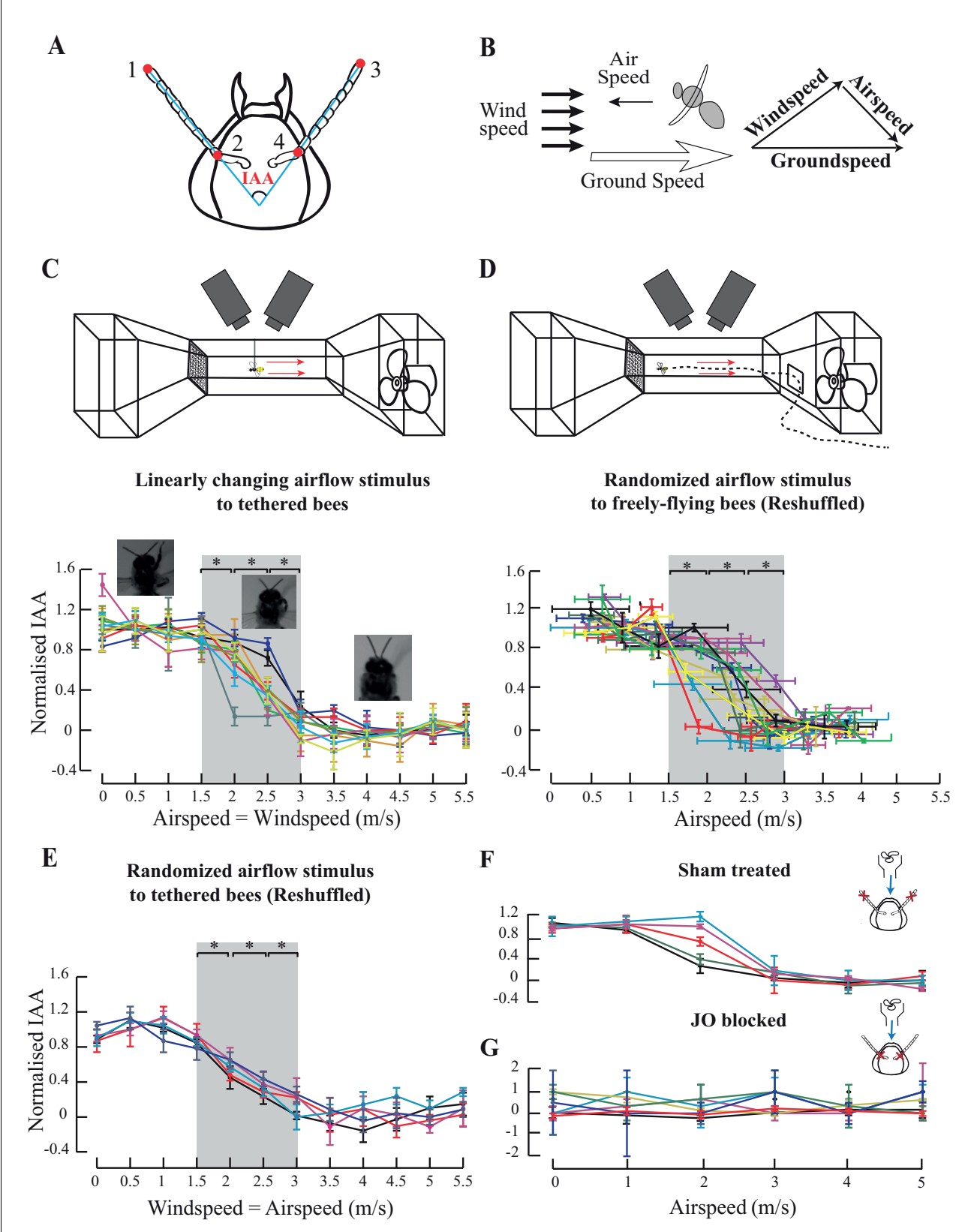

**Figure 1.** Antennal responses to changing airflow. (**A**) Inter-Antennal Angle (IAA) is measured by digitizing 4 points (red circles) on the antennae in all frames. (**B**) Groundspeed is obtained by tracking Point#2 (from **A**) which is static relative to head. It is the vector sum of bee-controlled airspeed and

*Figure 1 continued on next page*

*Figure 1 continued*

experimenter-controlled windspeed. (**C**) **Top panel**: Response to ambient airflow in tethered bees. Tethered bees were positioned at the centre of the wind-tunnel test section, and facing upwind and the windspeed was linearly varied from 0 to 5.5 m/s. Two high-speed cameras positioned dorsally and laterally filmed the bees at 500 fps. **Bottom panel:** Normalised IAA response as a function of airspeed (or windspeed). We normalized IAA values between 0 (defined as the mean of values between 0 and 1.5 m/s) and 1 (defined as the mean of values between 3 and 5.5 m/s). Between 0–1.5 m/s and 3–5.5 m/s, normalized IAA did not significantly change (p>0.05, Moore's test). Between 1.5 and 3 m/s, normalized IAA sigmoidally decreased with airspeed, changing with each step (*p<0.0001, Moore's test, N=10; each colour represents one individual) relative to the preceding and succeeding values. Non-normalized data in *Figure 1—figure supplement 1B*. Here and everywhere we have plotted the means, and the error bars indicate the standard deviation of the mean. (**D**) **Top panel:** IAA response of freely-flying bees to ambient airflow. Bees were trained to enter the wind tunnel through a side-door and fly upwind past the test section to a feeder. High-speed cameras placed and operated as in (**C**) filmed their IAA response. **Bottom panel:** Normalised IAA response as a function of airspeed in free flight. Between airspeeds of 1.5 to 3 m/s, IAA changed significantly (*p<0.0001, Moore's test, N=10) relative to the preceding and succeeding values, but saturated at airspeeds less than 1.5 m/s and greater than 3 m/s. Non-normalized data in *Figure 1—figure supplement 1D*. (**E**) IAA responses to random sequence of ambient airflow in tethered bees. We presented the bees with airflow values between 0 and 5.5 m/s in a random sequence and plotted the normalised IAA response as a function of airspeed values reshuffled to lie in increasing order. As in 1C, IAA sigmoidally decreased with airspeed, significantly changing between 1.5 and 3 m/s (*p<0.0001, Moore's test, N=5; each colour represents one individual.) From 0–1 m/s and 3.5–5.5 m/s, the normalized IAA did not significantly change. For non-normalized data, see *Figure 1—figure supplement 1C*. (**F** and **G** insets) Experiments with sham-treated and JO-restricted bees. Red crosses indicate the location of applied glue, and blue arrow the presence of airflow. (**F**) Normalised IAA vs. airspeed in sham-treated bees. Sham-treated bees (N=5) show responses similar to untreated bees (compare with *Figure 1C*). Each coloured line represents an individual bee (Non-normalized data in *Figure 1—figure supplement 1H*). Change in IAA (*Figure 1—figure supplement 1H*) is in the same range as untreated bees (compare with *Figure 1—figure supplement 1B*). (**G**) Bees with restricted JO do not respond to airspeed change. When the pedicel-flagellum joint is glued, IAA does not vary significantly with changing airspeed (*p>0.1, Moore's test, N=7). Each colour represents an individual (Non-normalized data in *Figure 1—figure supplement 1I*).

The following figure supplement is available for figure 1:

**Figure supplement 1.** Antennal responses to changing airflow.

increased, both antennae moved forward and mean IAA at each airflow value decreased as a sigmoidal function of airspeed (*Figure 1C,D*). The linear region of the sigmoid lay between 1.5 to 3 m/s (grey bar, *Figure 1C,D*), changing by approximately 35° (*Figure 1—figure supplement 1B*) in tethered and 25° (*Figure 1—figure supplement 1D*) in freely-flying bees, significantly decreasing with each step change in airspeed (step change = 0.5 m/s; *p<0.0001; Moore's paired test; N=10). This behaviour was similar regardless of whether airspeed stimulus was presented in linearly increasing or decreasing steps (*Figure 1C*, *Figure 1—figure supplement 1B*) or in random order (*Figure 1E*, *Figure 1—figure supplement 1C*). Thus, antennal position is calibrated against absolute airspeeds, independent of time history. At airspeeds less than 1.5 m/s or greater than 3 m/s, mean IAA at each airflow plateaued in both tethered (*Figure 1—figure supplement 1B,C*) and freely-flying bees (*Figure 1—figure supplement 1D*).

Previous studies on antennal responses of tethered bees to changing airflow did not report a zone of saturation (*Heran, 1957*) perhaps due to coarser sampling of their data (*Figure 1—figure supplement 1E*). We, however, consistently observed sigmoidal mean IAA responses, similar to locusts (*Gewecke, 1974*). As frontal airflow in the wind tunnel increased, freely-flying honeybees modulated their flight to maintain roughly constant groundspeed of *ca.* 0.4 m/s (*Figure 1—figure supplement 1F*; also [*Barron and Srinivasan, 2006*]). In these bees, we observed no correlation between groundspeed and IAA (*Figure 1—figure supplement 1G*).

## Johnston's organs mediate the antennal responses to airflow

Previous researchers have implicated the antennal mechanosensory Johnston's organs (JO) in sensing airflow cues (*Gewecke, 1974*; *Heran, 1957*; *Yorozu et al., 2009*). JO spans the pedicel-flagellar joint in the antennae of all Neopteran insects, and tracks the motion of the flagella relative to pedicel. In most insects, the JO consists of several hundred scolopidial units that are range-fractionated. These enable both exquisite sensitivity and narrowly-tuned sensing over a large range of stimulus frequencies (*Yorozu et al., 2009*; *Dieudonné et al., 2014*). Does JO also mediate the observed antennal response to changes in airspeed? To test this hypothesis, we attenuated JO feedback by gluing the pedicel-flagellar joint (see Materials and methods) in tethered honeybees, and measured their antennal responses to airflow. To control for the extraneous effects of glue on the antenna

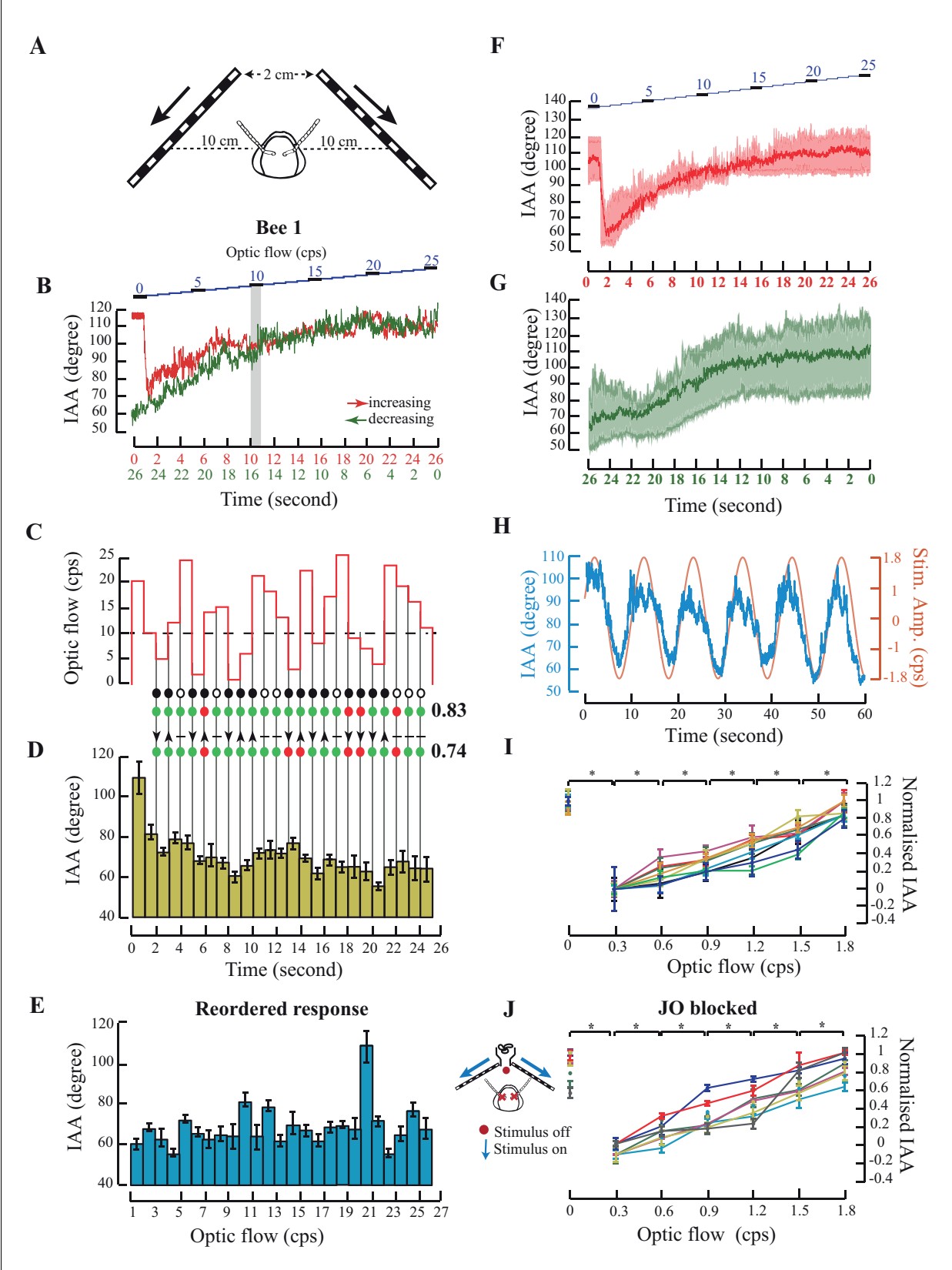

**Figure 2.** Antennal responses to changing temporal frequency of optic flow. (**A**) IAA response to optic flow in tethered bees. Bees were positioned central and at 10 cm from the two screens separated by 2 cm at the apex. (**B**) Sample IAA response to linearly increasing (red) or decreasing (green)
*Figure 2 continued on next page*

*Figure 2 continued*

optic flow stimulus. (Top panel) Stimulus comprises of a visual grating moving from front to back at temporal frequencies ranging between 0–25 cycles/s (cps), in steps of 1 cps. Each step lasts for 1 s (blue). The IAA response to optic flow saturates beyond the threshold of 10 cps (grey bar; also Materials and methods for threshold calculation). (C–E) IAA Response to randomized optic flow values. The above honeybee (the individual shown in *Figure 2B–E*, bee #1), was presented with randomized optic flow stimulus (C). The precise temporal sequence of randomized stimulus varied between bees. Dotted line shows the 10 cps threshold. Each 1 cps step lasts for 1 s from 1–25 cps. IAA response is plotted in two ways: the IAA response to the randomized stimulus (D, olive green bars), and the response reshuffled in increasing order of temporal frequencies (E; blue bars). The peak at 20 cps is due to the sharp IAA readjustment at stimulus onset. Grey lines between C and D indicate step transitions in temporal frequency values. IAA is predicted to change when stimulus changes occur below or across threshold. Predicted changes in IAA are marked by black circle and no change by white circle. IAA is predicted to decrease (down arrowhead) when optic flow changes from high-to-low under or across threshold, and increase (up arrowhead) when values change from low-to-high under or across threshold. Changes in temporal frequency above threshold (horizontal line) yield no antennal response. In both tests, correct predictions are marked by green and wrong predictions by red circles, and fraction of correct predictions vs. total number indicated beside each test. In this instance, we correctly predicted when IAA would change with 83% accuracy (19/23=0.83), and the direction of its change with 74% accuracy. (F, G) Summary figures showing the IAA response of three individuals (shown in *Figure 2B–E*, *Figure 2—figure supplement 1A–D and E–H*) (F) IAA responses for optic flow rates increasing linearly from 0 to 25 cps. The mean response from all three bees is shown in bold red and the spread of the data is shown in the background of the plot. (G) IAA responses for optic flow rates decreasing linearly from 25 to 0 cps. The time axis is shown in the opposite manner because the first optic flow rate that is presented to the tethered bee is 25 cps. The mean of the IAA responses from all three individuals is shown in bold green and the spread of the data shown in the background. (H) IAA response to sinusoidal moving visual gratings. Sinusoidal moving grating (orange, amplitude=1.8 cps; period =10 s) stimulus elicits correlated IAA responses (blue). (I) Normalized IAA response vs. optic flow rate between 0 and 1.8 cps. Each step increase of 0.3 cps in the optic flow rate elicits significant changes in IAA (*$p<0.05$, Moore's test, N=9). IAA values were normalized relative to maximum and minimum values for each individual (raw data in *Figure 2—figure supplement 2A*). (J, inset) IAA Responses of the bees with restricted Johnston's organs. Red crosses indicate glue location. To the set-up in *Figure 2A*, we added a ducted fan to provide collimated airflow. Red dot indicates absence of airflow, and blue arrows indicate presence of optic flow. (J) Normalized IAA response to changes in optic flow. IAA changes with change in optic flow (*$p<0.0001$, Moore's test, N=7). In *Figure 1G* and *2J*, the same individuals share color. Normalisation procedure is the same as in *Figure 2G* and the raw data has been shown in *Figure 2—figure supplement 2B*.

The following figure supplements are available for figure 2:

**Figure supplement 1.** Antennal responses to changing optic flow rates.

**Figure supplement 2.** Antennal responses to changing optic flow rates.

(e.g. due to added weight), we glued the second annulus of the flagella from the tip (*sham-treatment*) in a separate group of the bees. Application of glue at the JO does not affect the movement of the antennae because the pedicellar-flagellar joints have no muscles, and hence motion around these joints are passive. Unlike sham-treated bees which showed the typical sigmoid response to frontal airflow (*Figure 1F*, non-normalized data in *Figure 1—figure supplement 1H*, N=5), bees with restricted JO positioned their antennae at the onset of flight (*Figure 1—figure supplement 1I*; also [*Krishnan et al., 2012*]) but their mean IAA at each airflow was insensitive to changing airspeeds (*Figure 1G*, N=7; $p>0.1$, Moore's paired test). These experiments established that JO input is required for antennal positioning during flight, but not at flight onset (*Krishnan et al., 2012*). Thus, the antennal positioning at flight onset is a separate process from the inflight modulation of IAA.

## Bees' antennae move backward in response to increasing front-to-back optic flow

Flying insects rely heavily on optic flow for flight control, most notably during slower flight manoeuvres such as landing or hovering (*Baird et al., 2013*). For example, visual feedback is critical for regulation of groundspeed and height, or in centring trajectories through narrow corridors in diverse flying insects (bees: [*Srinivasan et al., 1996*; *Baird et al., 2005*], *Drosophila:* [*Straw et al., 2010*], moths: [*Kuenen and Baker, 1982*], review: [*Collett et al., 1993*]). It also influences antennal position in insects that antennate during walking (*Ye et al., 2003*; *Honegger, 1981*).

Under natural flight conditions, front-to-back optic flow stimulus accompanies frontal airflow stimulus. Does the antennomotor system also respond to optic flow in addition to airflow? To address this question, we simulated forward flight conditions by presenting tethered flying bees with changing temporal frequency of front-to-back optic flow cues on two LED screens, and measured their IAA (*Figure 2A*). Note that, previous studies on optic flow dependent behaviours in honeybees have

typically demonstrated that they extract angular velocity (ratio of temporal and spatial frequency) cues from the image motion, independent of spatial (*Srinivasan et al., 1991*) or temporal frequencies (*Baird et al., 2013*). In the experiments reported here, we have kept spatial frequency constant and only varied temporal frequency as the main experimental variable which means that the angular velocity is the temporal frequency times some multiplication factor.

In a single flight bout (defined as an uninterrupted session between initiation and cessation of flight), tethered bees first experienced a static screen of black-and-white gratings (i.e. temporal frequency = 0) followed by a sequence of gratings moving from 0 to 25 cycle/s (cps) in discrete steps of 1 cps to simulate increasing flight speed (red curve, *Figure 2B*; *Figure 2—figure supplement 1A, E*; summary figure showing data from all individuals, *Figure 2F*). In the subsequent flight bout, the temporal frequency decreased stepwise from 25 to 0 cps (green curve, *Figure 2B*; *Figure 2—figure supplement 1A,E*; summary figure showing data from all individuals, *Figure 2G*) to simulate decreasing flight speed. Each stimulus step lasted for 1 s and the entire protocol for 26 s. When temporal frequency increased from 0 to 25 cps, an initial sharp adjustment in IAA (red curve, *Figure 2B*, summary figure *Figure 2F*) was followed by a graded response between 0 to approx. 10 cps (grey bar, *Figure 2B*), beyond which the value reached a plateau. The response curve for linearly decreasing optic flow (25 to 0 cps, green curve, *Figure 2B*, summary figure *Figure 2G*) was similar to the increasing cues, except for the lack of initial sharp IAA adjustment from zero to non-zero optic flow (also *Figure 2—figure supplement 1A,E*).

In these experiments, the optic flow was patterned to monotonically increase or decrease. Graded IAA responses to such optic flow patterns suggested two possibilities.

First, as in the case of airflow-based response, IAA response curve is innately calibrated against specific temporal frequency values, independent of its time history. If true, the response curve should remain invariant when the temporal frequency of the stimulus is presented in random order. This is an unlikely possibility because absolute values of optic flow are meaningless without prior knowledge of the spatial structure of the world.

Second, that bees respond to *changes in* temporal frequency rather than their absolute value. If true, IAA would increase (*or decrease*) in response to positive (*or negative*) changes in temporal frequency within the operating range (e.g. 0–10 cps for the bee in *Figure 2B–E*) while remaining unchanged outside this range (*i.e.* >10 cps). The threshold value was calculated separately for each bee (see Materials and methods for details). The latter criteria lead to specific predictions of when to expect changes in IAA (filled black circle; *Figure 2C,D*) and in which direction (arrow; *Figure 2C,D*).

To test the above hypotheses, we presented tethered honeybees with grating speeds between 1 to 25 cps in randomized order (*Figure 2C–E* and *Figure 2—figure supplement 1B–D*, *Figure 2—figure supplement 1F–H*) and compared the measured vs. predicted IAA response (*Figure 2D*; also *Figure 2—figure supplement 1C,G*). For each bee, we scored these as the fraction of correct predictions; a score of 1 corresponds to all correct, and 0 to all incorrect predictions. In all the cases, the total prediction score matched the actual observed score well above chance levels (*p<0.001; Student's T test; see Materials and methods for details). When stimulus bins were rearranged in ascending order of temporal frequency, the reshuffled mean IAA response was not graded (compare *Figure 2B and E*; *Figure 2—figure supplement 1A and D*; *Figure 2—figure supplement 1E and H*) ruling out the possibility of innate calibration of IAA against optic flow. Thus, our data show that antennomotor activity in honey bees tracks *changes in* temporal frequency, and not their absolute value.

Bees typically experience the steepest gradients in optic flow during landing or hovering over flowers when optic flow rates are low (*Baird et al., 2013*). Hence, in the follow-up experiments, we focused on the lower range of stimuli from 0–1.8 cps (*Figure 2H–J*). Even in this narrow range, IAA responses tracked the magnitude and direction of grating patterns on the screen. For example, bee antennae robustly tracked simple sinusoidal stimuli that were both, cycled between front-to-back (0 to 1.8 cps), and back-to-front grating movement (0 to -1.8 to 0 cps) over the 10 s duration (*Figure 2H*), thus verifying that the antennae move in both directions as a function of optic flow. To characterize their responses in this narrow range of temporal frequency (0–1.8 cps), the tethered bees initiated wing flapping in front of a blank screen, followed by a sequence of black-and-white gratings moving front-to-back from 1.8 to 0.3 cps. Each grating sequence lasted for 6 s, interspersed with a static screen (zero optic flow) for 3 s. Despite inter-animal variability in the set points for IAA at zero optic flow (*Figure 2—figure supplement 2A*), all bees increased their IAA as a function of

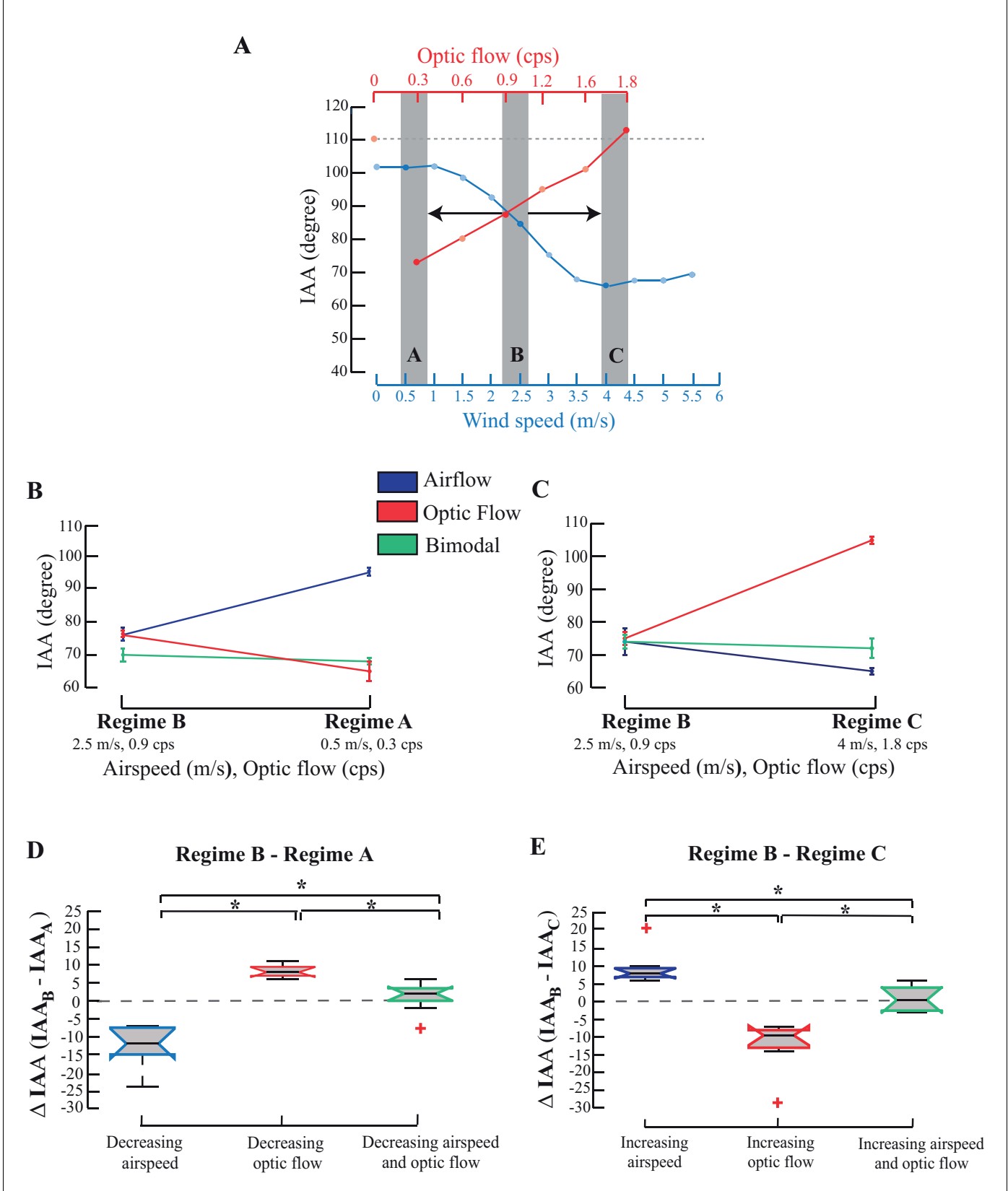

**Figure 3.** IAA responses to combinatorial stimuli. (**A**) IAA response curve of a tethered bee to changing airflow (blue) and optic flow (red). Regime A represents the combination of low optic flow rate and airspeed (temporal frequency=0.3 cps; airspeed=0.5 m/s); Regime B represents combination of

*Figure 3 continued on next page*

*Figure 3 continued*

intermediate optic flow rate and airspeed (temporal frequency =0.9 cps; airspeed=2.5 m/s), and Regime C represents high optic flow rate combined and airspeed (temporal frequency =1.8 cps, airspeed=4 m/s). The dashed line (grey) at the top of the plot indicates the IAA response at 0 optic flow. (B, C) Representative data from a single individual when values transition from Regime B→ A (B) or from Regime B→C (C). IAA responses to airspeed (blue), and optic flow (red), and combination of airspeed and optic flow (green) plot. (D, E) ΔIAA response to step changes in airspeed, optic flow or combined cues. We have represented the mean ΔIAA values as notched plots. The extent of box shows the inter-quartile range and the lower and upper bounds of the box represent the 25th and the 75th quartiles. The line in the box represents the median of the data and there is a 'notch' around this median for easy comparison of the notched boxes with each other. If the notches of two boxes do not overlap, their medians are statistically significantly different from each other. The red crosses indicate the outliers in the data. The whiskers extend to the most extreme data point that is not considered to be an outlier. * represents statistically significant difference (*p<0.001, Moore's test, N=8). Mean ΔIAA is significantly different from a hypothetical mean of zero (*p<0.05, ANOVA; post hoc Tukey's HSD test, N=8) when only airspeed or optic flow are varied, but not when a stimulus combination is co-varied (*p>0.5, ANOVA; post hoc Tukey's HSD test, N=8).

optic flow (*Figure 2I*), with each step change of 0.3 cps eliciting significant increase in mean IAA (*p<0.0001; Moore's paired test, N=10) at that optic flow rate. The IAA response of tethered bees to optic flow remained intact (*Figure 2J*; *Figure 2—figure supplement 2B*) even when their JO were restricted by gluing the pedicel-flagellar joint, establishing that it is independent of the JO pathway. Thus, front-to-back optic flow causes antenna to move backward, opposite to frontal airflow which causes antenna to move actively forward.

## Visual and airflow cues combine to influence antennal positioning

The experiments outlined above established that inputs from two sensory modalities independently helped the bees to sense airspeed and optic flow. Whereas the JO-based antennal mechanosensory feedback was calibrated against absolute values of airspeed, visual feedback was calibrated relative to changes in optic flow under or across a threshold value. These experiments required flying bees to respond to single sensory cues at a time. Hence, we next measured IAA response of tethered bees to simultaneous visual and airflow cues. The paired stimuli combinations were arbitrarily drawn from values in previous experiments, and are not naturally correlated.

How do multimodal cues affect IAA response when simultaneously presented? Optic flow (red, *Figure 3A*) and airflow (blue, *Figure 3A*) cues were presented in three combinations, represented by three regimes A, B or C (grey bars; *Figure 3A*). Responses of a single bee are shown in *Figure 3B,C*. In each trial, we started the experiment with an intermediate bimodal combination (grey bars, middle Regime B; optic flow=0.9 cps, airspeed =2.5 m/s) and in random order, either decreased (to Regime A; optic flow =0.3 cps, airspeed =0.5 m/s; *Figure 3B*) or increased these values (to Regime C, optic flow =1.8 cps, airspeed =4 m/s; *Figure 3C*). Antennae maintained position when the cue combination changed from Regime B to Regime A (green line; *Figure 3B*; Pp>0.1, Moore's test), but not when only airflow or optic flow was changed. When airspeed alone (i.e. with static grating) decreased from 2.5 to 0.5 m/s (blue line; *Figure 3B*; p<0.001, Moore's test), both antennae moved backward. However, they moved forward when optic flow alone (i.e. with fan off) decreased from 0.9 to 0.3 cps (red line; *Figure 3B*; p<0.001; Moore's test). Similarly, when the cue combination changed from Regime B to Regime C, antennae again maintained position (green line; *Figure 3C*; p>0.1, Moore's test), but moved forward when airflow alone increased from 2.5 to 4 m/s (blue line; *Figure 3C*) and backward when optic flow alone increased from 0.9 to 1.8 cps (red line; *Figure 3C*). Thus, visual and airflow cues elicit opposite IAA responses which, when acting in concert, maintain antennal position.

This was also borne out in the pooled data over multiple trials (*Figure 3D,E*). Decrease of only airspeed resulted in a negative mean ΔIAA (blue, *Figure 3D*; *p<0.001; Moore's test), whereas increase resulted in a positive mean ΔIAA (blue, *Figure 3E*; *p<0.001; Moore's test). Similarly, a decrease in only optic flow resulted in positive mean ΔIAA values, but its increase led to negative mean ΔIAA (red, compare *Figure 3D,E*; *p<0.001; Moore's test) compared to the IAA values in Regime B. Here, mean ΔIAA values significantly differed from a hypothetical mean of zero (one-way ANOVA with post-hoc Tukey's Honest Significant Difference test, *p<0.05). However, when both cues were co-varied, ΔIAA did not significantly change (green, *Figure 3D,E*; p>0.1, Moore's test) and their means were statistically indistinguishable from zero.

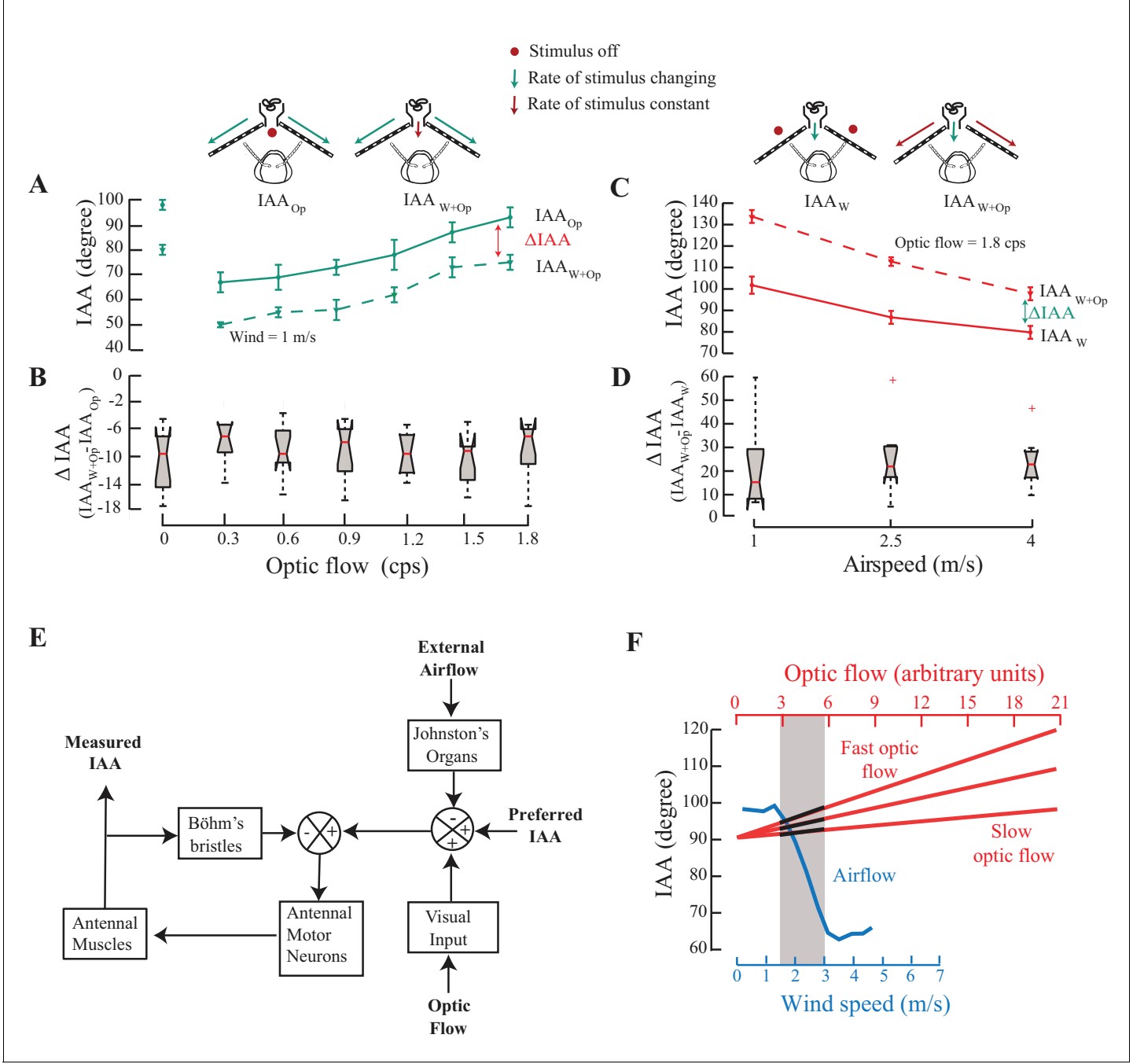

**Figure 4.** IAA responses to changes in optic flow in presence of constant airflow (A, B), and to changes in airspeed in presence of constant optic flow cues (C, D). (A) IAA responses to optic flow rates varying from 0 to 1.8 cps in presence of still air (IAA$_{Op}$; solid green line vs. steady 1 m/s airflow (IAA$_{W+Op}$; dotted green line). Mean difference between IAA$_{Op}$ and IAA$_{W+Op}$ is significant at each optic flow rate value (*p<0.0001, Moore's test, N=8). (B) We have represented the mean ΔIAA values as notched plots. The range of ΔIAA (=IAA$_{W+Op}$ - IAA$_{Op}$) is not significantly different over various optic flow values. Each mean ΔIAA is significantly different from a hypothetical mean of 0 (*p<0.05, ANOVA and post hoc Tukey's HSD test, N=8). (C) IAA responses to varying airspeed at 1, 2.5 and 4 m/s in presence of no optic flow (IAA$_W$; solid red line; top left panel) vs. steady optic flow of 1.8 cps (IAA$_{W+Op}$; dotted red line; top right panel). Again, mean IAA values at each rate of optic flow are significantly different (*p<0.0001, Moore's test, N=9) for the two cases. (D) The range of ΔIAA plotted as notched plots are again not significantly different across the various airspeeds, but each mean ΔIAA value is significantly different from a hypothetical mean of 0 (*p<0.05, N=9, ANOVA and post hoc Tukey's HSD test). (E) A general model of the antennal positioning response to airspeed and optic flow cues, including the role of mechanosensory hair plates (Böhm's bristles) in antennal positioning response. (F) The crossmodal calibration hypothesis proposes that insects simultaneously sample airflow and optic flow, and use response characteristics of airflow sensing to calibrate optic flow. Determining how the sampled optic flow varies in the dynamic range of airspeeds enables insects to linearly extrapolate the optic flow response curve over a greater range. The grey bar represents the stimulus range in which such

*Figure 4 continued on next page*

*Figure 4 continued*

simultaneous airflow (blue) and optic flow (black on red) measurements are made. According to this hypothesis, once a specific airflow value is correlated against the observed optic flow, which could be slow or fast, it can then be used to make measurements over much greater range of airspeeds.

How does the presence of one cue alter the antennal response curve of the other cue? We considered three possibilities. First, if the response curve of Cue A remains unaltered in presence of a constant Cue B, then it is likely that the system adapts to cue B. Second, if the response curve to Cue A is uniformly offset by the presence of constant Cue B, then the cross-modal influence is likely linear and summative. Third, if response curve to Cue A is non-uniformly offset in presence of constant Cue B, then crossmodal influences are likely to be non-linear. Testing for possibilities provides insights into how inputs from different modalities combine to determine the antennal position.

We measured IAA against optic flow in presence of two constant values (0 and 1 m/s) of airspeed. The two curves were uniformly offset over the range from 0 to 1.8 cps (compare dotted and solid lines, *Figure 4A,B*; *p<0.05, Moore's test, N=8). Similarly, for airspeeds between 0 and 4 m/s, we observed a steady offset for greater optic flow, but in the opposite direction (compare dotted and solid lines, *Figure 4C,D*; *p<0.05, Moore's test, N=9). In both experiments, the presence of constant optic flow (*or airflow*) altered the set point of the antenna to a new mean value that is greater (*or less*) than the original set point (*Figure 4B,D*) and the antenna responded to variation in airflow (*or optic flow*) relative to the new set point. Thus, the antennomotor system linearly combines the multimodal cues and recalibrates accordingly.

## Discussion

### Antennal positioning responses involve multimodal sensory integration

The above study provides us with insights into how the insect antennomotor system integrates the multiple sensory cues that it encounters during flight. This can be summarized in a general schematic of the multi-modal integration of antennal positioning behaviour in flying honeybees (*Figure 4E*). For the sake of completeness, this model must also include the role of the mechanosensory hair plates (or Böhm's bristles) in reflexive antennal positioning, which appears conserved across insects according to our studies in moths (*Krishnan et al., 2012*), bees and crickets (*Sant and Sane, 2016*). To summarize these studies: antennal hair plates are stimulated when the antenna undergoes a substantial movement in the scape-head capsule or pedicel-scape joints. The mechanosensory neurons underlying these bristles directly project into the Antennal Motor and Mechanosensory Center (AMMC) where they arborize on the dendritic fields of antennal motor neurons and activate them. Any gross change in antennal position thus elicits a rapid reflexive correction of the antennal position with latencies under 10 ms in hawk moths (*Krishnan et al., 2012*) and probably on the same order in honeybees and other insects. Based on these data, we proposed a model to describe the antennal positioning reflex loop mediated *via* antennal mechanosensors (*Krishnan et al., 2012*). Although arrangement of the hair plates in different insects varies, the underlying neural circuitry and hair plate function is conserved in moths, bees, and most other insect orders (*Krishnan and Sane, 2015*). Previous studies in hawk moths have described only how the antenna, once positioned, reflexively maintains this position (*Krishnan et al., 2012*), and also their response to visual cues (*Krishnan and Sane, 2014*). However, the combinatorial role of these cues in antennal positioning behaviour *during flight* remained unclear.

Here, we show that the reflexive maintenance of in-flight antennal positioning is simultaneously modulated by both visual (from eyes) and airflow (from JO) feedback during flight. This study also shows that their action is *via* multi-modal pathways which combine in a mutually antagonistic fashion; whereas frontal airflow detected by JO reduces IAA, the front-to-back optic flow detected by compound eyes increases IAA. The combination of these inputs modifies IAA to a new set point, which is then maintained by the antennal hair plates *via* a negative feedback loop that ensures rapid maintenance and correction of the antennal position. The block diagram in *Figure 4E* thus provides a model for multimodal sensory control of antennal positioning response. It is possible that other

inputs, such as olfaction, also additionally modulate the antennomotor responses, which is the subject of future studies.

## General relevance of antennal positioning response to mechanosensory and visual cues

Our study shows that the initial positioning of antennae at the flight onset (*antennal deployment*) is separate from its later inflight maintenance by the multimodal inputs (*inflight antennal positioning*). Both behaviours require the hair plate-mediated reflex pathway but the set-point of this reflex system is under multimodal control in the latter case. In particular, restriction of the JO feedback has no effect on the *antennal deployment behaviour*, whereas it completely disrupts *inflight antennal positioning*. It has been suggested that inflight modulation of IAA maintains the scolopidial units of the JO in their operating range (*Hinterwirth et al., 2012*). Although this hypothesis remains unaddressed and is beyond the scope of this paper, our data clearly show that inflight antennal movements are precisely modulated, and product of both visual (from compound eyes) and mechanosensory (from the hair plates and JO) input. Visual feedback also induces directionally-sensitive antennal movements in other insects such as hawk moths (*Krishnan and Sane, 2014*), *Drosophila* (*Mamiya et al., 2011*), and many orthopteran insects (*Honegger, 1981*; *Ye et al., 2003*). Unlike the hair plate reflexes which are strictly unilateral (*Krishnan et al., 2012*), visual feedback drives the activity of both ipsi- and contralateral antennal motor neurons in moths (*Krishnan and Sane, 2014*) and may therefore serve to coordinate the movements of both antennae. The mechanosensory pathways mediated by the hair plates then provide rapid reflexive, local correction of the intended position of each antenna.

Recent studies have also described how the integration of antennal mechanosensory and visual inputs mediates abdominal flexion in moths (*Hinterwirth and Daniel, 2010*) which in turn is relevant for flight control and balance. In honeybees too, their combined effect on the abdominal streamlining behaviour enables drag reduction during flight (*Luu et al., 2011*; *Taylor et al., 2013*). In freely-flying moths, the stimulation of antennal muscles elicits abdominal flexion accompanied by a change in flight trajectory (*Hinterwirth et al., 2012*). Similarly, integration of mechanosensory and visual information mediates flight control in other insects such as *Drosophila* (*Sherman and Dickinson, 2003*; *Fuller et al., 2014*). Whereas the relative influence of mechanosensory input is greater during rapid turns, visual input is more important during slower rotations (*Sherman and Dickinson, 2003*; *2004*).

The integration of these multimodal cues quite likely occurs in the Antennal Motor and Mechanosensory Centre (AMMC) region of the insect brain, which houses the soma of the antennal motor neurons. This region also receives the arbors of motion-sensitive visual interneurons (*Hertel and Maronde, 1987*), and likely also the inputs from cephalic hair mechano-afferents. These different modalities are known to influence diverse flight behaviours. It is thus likely that descending interneurons transduce multimodal sensory information that elicits these behaviours. In locusts, the tritocerebral commissure giant (TCG) interneurone integrates visual and airflow information (*Bacon and Tyrer, 1978*), while other descending interneurons in crickets integrate antennal mechanosensory and visual inputs (*Gebhardt and Honegger, 2001*). The importance of AMMC as the site of integration of multimodal sensory feedback is an important topic of future study.

## A putative mechanism of calibration of optic flow based speedometer in honeybees.

The data on antennal position are also relevant to studies on speedometry in honeybees. A key finding of this paper is that IAA responds to absolute airspeed values from 1.5 to 3 m/s, which is narrow compared to the typical range of airspeeds (between 0 and 7 m/s; [*Wenner, 1963*])of honeybees in free flight. IAA is also calibrated against optic flow cues, but this feedback depends on the spatial structure of the environment. As shown in studies over the past two decades, the odometer in honeybees is quite clearly visual (*Srinivasan, 2014*). The visual system can extract the overall image motion, which is perceived by and integrated over the motion detectors in the retina (*Zanker et al., 1999*). Honeybees trained to find food at a specific distance in tunnels lined with visual stripes can correctly judge distances even when the spatial frequency of the stripes is altered (*Baird et al., 2005*; *Srinivasan et al., 1996*; *1997*; *2000*; *2014*). Moreover, they estimate distances correctly

despite head or tailwinds, suggesting that parameters such as windspeed, flight duration and number of wing beats etc. play no role in the odometry. Indeed, image motion cues can also be used to estimate depth and distinguish between near and far objects independent of their size (*Kirchner and Srinivasan, 1989*; *Srinivasan et al., 1989*; *Zhang et al., 1992*). Thus, bees appear to be able to separately process size and velocity cues (*Srinivasan et al., 1993*).

Complement to the visual odometer, we show here that honeybees are also able to extract information about both airspeed and groundspeed. Importantly, the airspeed calibration mediated by JO is absolute, but works over a narrow speed range, whereas the groundspeed calibration mediated by optic flow can be tuned over a broader range. We propose the hypothesis that honeybees calibrate their optic flow speedometer using their JO-based system that measures absolute airspeeds (*Figure 4F*). At the onset of a regular flight bout, bees simultaneously sample the optic flow (red lines) and airspeed (blue line). The three red lines correspond to temporal frequency of optic flow that is slow (i.e. if objects are far away) to fast (i.e. if objects are close). Sampling the optic flow (*Figure 4F*; black segments on the red lines) corresponding to the dynamic range of the JO-based airspeed sensor (e.g. between ~1.5 to 3 m/s) provides bees with the means to cross-calibrate average optic flow against absolute airspeeds. As the airspeed-based estimation only works in a narrow range from 1.5 to 3 m/s, the optic flow-based speedometer requires extrapolation well beyond the range of the antennal system. As shown in *Figure 4A–D*, the presence of one cue linearly offsets the response curve for the other, analogous to using a standardized *albeit* narrow-ranging measure (e.g. airflow-based speedometer) to calibrate another arbitrarily scalable measure (e.g. the optic flow based speedometer). A disadvantage of such a system is that it must be recalibrated at the beginning of each bout, or within bouts in case of sudden gusts of wind which may confound the calibration. Under such circumstances, we predict that honeybees would have to slow down to recalibrate their optic flow based speedometer against the airspeed-based speedometer, before resuming flight.

Our data throw some light on puzzling observations from previous studies. In their study on visual regulation of groundspeeds in honeybees, Barron and Srinivasan (*Barron and Srinivasan, 2006*) observed that honeybees freely flying within a tunnel lined with checkerboard patterns flew at speeds of approximately 0.4 m/s. However, when the checkerboard pattern was replaced by an axial-stripe pattern which offered sparse optic flow cues, its groundspeed was maintained with low variance at a value of 1.4 m/s. Maintenance of ground speeds (which equals airspeed in still air) requires that bees be able to 'sense' their speed even when optic flow cues are sparse. How is this possible if speed is maintained only *via* optic flow? To explain this result, we propose that, in absence of optic flow, the bees rely on their JO-based airflow sensors to both set *and maintain* groundspeed. It is worth noting that the value of 1.4 m/s lies at the cusp of the airflow response curve (*Figure 1C*), and is maintained as the new reference at different wind speeds in sparse optic flow (see *Figure 3* in *Barron and Srinivasan [2006]*).

A JO-based antennal airflow sensing, can in principle offer a 'true' airspeed measure for odometry. It is not presently clear if the visual odometer would need such information as the current model for visual odometry is able to quite robustly explain most observations. One would expect such a speedometer system to accumulate errors as the range increases, as has also been observed with the honeybee odometer, which follows the Weber's law (*Cheng et al., 1999*). Future studies will be required to determine if the several predictions that emerge from this model about honeybee speed control hold true. The antennal positioning responses thus provide unique insights into how honeybees and perhaps other insects sense and combine information from multiple modalities to help reduce ambiguities arising from drift in any one modality.

## Materials and methods

All experiments were performed on the European honeybees, *Apis mellifera*, maintained in an apiary in an outdoor insectary approximately 60 m from the lab. For laboratory-based experiments, we either captured individual forager bees or trained them to fly to the wind tunnel for a food reward. In free-flight experiments, trained bees were marked with acrylic paint on the thorax and abdomen for easy identification of individuals. In experiments involving purely airflow cues, the experimental manipulations were performed on both tethered and freely-flying bees. Experiments involving purely

visual cues as well as combined visual and air flow cues were performed exclusively on tethered bees.

## Tethering protocol

After capturing individual forager bees at the hive, we cold-anaesthetized them on ice until they were inactive. The anaesthetized bees were then dorsally tethered to a bent metal rod (30 mm in length and 0.2 mm diameter) using a synthetic rubber based adhesive (Fevibond, Pidilite, Mumbai, India). The bee was then provided with sucrose solution and left to recover for 45 min. When performing experiments involving visual stimuli, we dark-adapted the bees in a darkened box during this recovery period. To elicit flight, we provided the bee with a piece of tissue paper to hold, and then suddenly withdrew it to elicit flight due to tarsal reflex.

## Airflow stimuli

We constructed an open-circuit wind tunnel (30 cm x 30 cm cross-section and 120 cm in length, with a 40 cm x 30 cm x 30 cm test section) with a fan, driven by a motor drive at one end that drew air through the wind tunnel. The wind tunnel was calibrated using a constant temperature mini-anemometer (Kurz 490S, Kurz Instruments, Inc., Monterey, CA), which was modified to take direct voltage readings (*Sane and Jacobson, 2006*). The voltage readings were consistently reproducible between 0–6 m/s at a step size of 0.5 m/s. To minimise confounding visual cues, the floor and walls of the wind tunnel were covered with white paper.

To determine how antennal position varies with ambient airflow, we provided tethered and freely flying bees with frontal airflow between 0 and 5.5 m/s and between 0 and 3.5 m/s in steps of 0.5 m/s respectively. At their fastest, freely flying honey bees have been estimated to fly at ~7.5 m/s in natural outdoor environments. Presence of headwinds or tailwinds of about 3 m/s does not affect the speed of the bees (*Wenner, 1963*) in their natural flight. In both cases, we filmed bees from the top and side with two synchronised high-speed cameras and calculated their IAA. We performed both tethered and free flight assays in the wind tunnel to look at how the antennae respond to changes in airflow.

## Visual stimulus

To investigate how visual motion influences the antennal positioning response, we modified a MATLAB demo routine from the Psychophysics toolbox (Mathworks Inc, Natick, MA) (*Brainard, 1997*; *Kleiner and Pelli, 2007*) to generate a moving visual grating pattern. The patterns were displayed on two LED monitors (Beetel 8"x14", 640x480 resolution, 60 Hz refresh rate) and controlled simultaneously by a master computer equipped with a dual-VGA graphics card (nVIDIA GeForce 9800GT). These screens did not show noticeable flicker at 1000 fps, as determined by filming them with a high-speed camera. The monitors were placed in a V configuration, with a 2 cm gap between them. The tethered bee was positioned at the centre of the two monitors, facing the apex, at a linear distance of 10 cm from each monitor (*Figure 2A*). The floor below the screens was layered with white paper to increase the contrast of the antennae during filming.

We provided open-loop forward translational stimuli (simulating front-to-back movement of the visual field) to the tethered bees, at a spatial frequency of 0.44 cm$^{-1}$ and various temporal frequencies (0 to 25 cps for the first set of the experiments and 1.8, 1.5, 1.2, 0.9 and 0.3 cps for the second set of experiments), corresponding approximately to angular speeds of 0 to 300 deg/s and 18, 15, 12, 9, 6 and 3 deg/s respectively. Using two synchronized high-speed video cameras (Phantom v7.3, Vision Research, Inc. Wayne, NJ), we filmed tethered flying bees at 250 fps, from both a side view and an overhead view. A third camera, synchronized to the first two, filmed the visual stimulus being displayed to the bee. We provided each bee with a range of temporal frequencies over a single flight bout, and digitized and analysed the whole video, the entire flight bout of the bee in that trial for the first set of experiments. In the second set of experiments in which we gave the bees optic flow from 0 to 1.8 cps, we digitised and analysed 50 frames of video per temporal frequency for 50 wing beats. In our analysis, we discarded bees that did not fly continuously through the entire protocol.

## Combination of visual and airflow stimuli

To provide both visual and airflow stimuli to the tethered bees, we placed a 4 wire DC fan in front of the bee in a 2 cm gap between the two screens in the set up described above, such that the tethered bee received frontal airflow (*Figure 2A*). The airflow was collimated and fairly laminar, and we ensured that there were no other sources of airflow. The speed of the fan was adjustable and we could measure the speed of the frontal airflow near the head of the bee using a hot wire anemometer. The fan was placed inside a rectangular box with a plexiglass tunnel in front of it that opened between the two screens so that the tethered bee received collimated frontal airflow.

## Experiments

### Antennal responses to change in airflow or optic cues

#### Response of tethered bees to changes in the speed of airflow

We suspended tethered bees from the ceiling of the wind-tunnel at the center of its test section, such that they faced upwind (*Figure 1C*, top). We then altered the speeds of external airflow using two protocols: a linear sequence of airflows (*Figure 1C*, *Figure 1—figure supplement 1B*) and a randomly generated sequence of windspeeds between 0 m/s and 5.5 m/s (*Figure 1E*, *Figure 1—figure supplement 1C*), and filmed the responses of their antennae at 500 fps from the top and the front using two synchronized high-speed video cameras. We digitized and analyzed 500 frames of video (or 250 wing beats) for each bee at each speed of airflow.

#### Responses of freely flying bees to airspeed changes

Using a 30% sugar solution in a yellow feeder as a reward, we trained foraging honey bees to fly into the laboratory from an outdoor insectary. The feeder was moved in a series of discrete steps from the insectary to the laboratory. The protocol typically took 7 days at the end of which the bees were trained to enter the wind tunnel and fly upwind to the feeder placed within the wind tunnel. To enable identification of individual bees, we marked them at the feeder with colored acrylic paint on the thorax and abdomen. A marked bee entering the wind tunnel experienced frontal external air flow at speeds ranging between 0 m/s and 3.5 m/s (step size of 0.5 m/s). We filmed the marked bees with two synchronized high speed cameras at 500 fps (*Figure 1D*, top). We obtained a 1 s video of flight in a straight trajectory for each bee at each wind speed value. The marked bee was filmed at a new randomly chosen airflow speed each time it entered the wind tunnel until the dataset for each bee was complete.

#### Responses of tethered bees to changes in the speed of optic flow

To determine the role of optic flow in antennal position, we placed tethered bees at the centre of a visual arena (*Figure 2A*) and varied the speed of the visual gratings to simulate front-to-back optic flow between 0 and 25 cps (corresponding to angular speeds, calculated as linear grating speed per unit distance from the eye of 0 to 300 deg/s). The spatial frequency of the grating was 0.44 cm$^{-1}$. In the first set of experiments, the bees were provided with a linear change of temporal frequency from 0 to 25 cps in both ascending and descending manner (*Figure 2F,G*). We also provided the bees with this range of optic flow (between 1–25 cps) but in a randomized manner (*Figure 2C–E*, *Figure 2—figure supplement 1B–D* and *Figure 2—figure supplement 1F–H*). In these protocols, each stimulus lasted for 1 s and the entire protocol lasted for 26 s in total. In the experiment in which we displayed sinusoidal rates of optic flow change, the optic flow was changed between 1.8 cps and -1.8 cps. Each sinewave lasted for 10 s and hence the entire trial for 6 sine waves lasted for 60 s (*Figure 2H*). In the second set of experiments when the optic flow changed between 0 and 1.8 cps, we displayed each grating speed for 6 s, followed by a static screen which was displayed for 3 s before the next grating speed value was displayed. The entire protocol lasted for 54 s in each experiment. We filmed the antennal positioning response of the bee with three synchronized high-speed cameras such that two of them were focused on the bee from the top and side, and a third on the computer monitor displaying the stimulus. We calculated the IAA and then determined the threshold of the IAA response for each bee separately.

## Antennal responses to diminished input from the Johnston's organs

To restrict input to the Johnston's Organs, we placed cold-anaesthetized bees on a chilled metal platform and applied cyanoacrylate adhesive to the pedicellar-flagellar joint of both antennae. After its recovery from anaesthesia, the tethered bee was placed at the centre of a visual arena comprising two monitors angled perpendicularly to each other, which displayed moving visual gratings (*Figure 2A*) to the bee. The bee was at a distance of 10 cm from each screen (*experimental bee*). We then tethered the bee and positioned it between the two monitors, with the DC fan at the apex (*Figure 1F,G* and *2J*). To control for the effects of the glue, in some bees we applied glue on the 2nd annulus of the flagella (*sham-treated bee*). In both *experimental* and *sham-treated bees*, we measured the response of the antennae to airflow and optic flow, in randomized order.

## Antennal responses to changing airflow and optic flow cues
### Changing airflow and optic flow cues simultaneously

To measure the antennal response in tethered bees to co-varying optic and airflow cues, we gave the bees stimuli-combinations from three regimes in their response range (*Figure 3A*): Regime A (airflow= 0.5 m/s; optic flow=0.3 cps); Regime B (airflow= 2.5 m/s; optic flow=0.9 cps) and Regime C (airflow= 4 m/s; optic flow=1.8 cps). The combination of cues was chosen from three regimes as described in the main text. The highest optic flow value was paired with the highest airflow value (Regime C), the mid-range optic flow value with the mid-range airflow value (Regime B) and the lowest optic flow with the lowest airflow (Regime A). We first filmed the antennae of bees flying at cues from Regime B and then either increased the speeds of airflow and optic flow to Regime C, or decreased the speeds to Regime A respectively. The order of these trials was randomized. The entire set of measurements was taken over a single flight bout, each pairing of stimulus lasting for 6 s with 10 s intervals in between. During this interval, a static grating was displayed on the screen while the airflow was being adjusted to the next value. As controls, we filmed the same bees under conditions of change in just one stimulus (changing airflow and not optic flow and changing optic flow but not airflow) to compare these with the responses to changes in both airflow and optic flow cues.

### Varying optic flow rates in the presence of a constant airflow

This experiment was performed on the same bees as the experiment on antennal responses to optic flow cues alone (from *Figure 2G*). The order in which these two experiments were carried out was randomized. In this case, optic flow was varied in the presence of a constant air flow of 1 m/s (*Figure 4A,B*).

### Varying airflow speeds in the presence of a constant optic flow

In a separate experiment on a different set of bees, we also investigated the effects of a constant optic flow on the antennal response to changes in airflow (*Figure 4C,D*). Tethered flying bees were provided with air flow speeds of 1 m/s, 2.5 m/s and 4 m/s, both in the absence and presence of a constant optic flow of 1.8 cps. The order of trials was randomized.

### Digitization of antennae and data analysis

We calibrated our high-speed cameras before and after each experiment. Using custom MATLAB software with a DLT (direct linear transformation) algorithm (details of the software can be found in [*Hedrick (2008)*])we digitized the antennal tips and bases of flying honey bees to reconstruct their positions in three dimensions (*Figure 1A*). We used these positions to calculate the IAA . The variation in antennal length in the digitized videos was used to estimate digitization error, which was within 0.5%–1.5% for tethered bees and 3%–5% for freely flying bees.

   We normalized the IAA data between 0 (defined as the mean of values between 0 and 1.5 m/s) and 1 (defined as the mean of values between 3 and 5 m/s). Between 0–1.5 m/s and 3–5 m/s, the normalized IAA did not significantly change ($p>0.05$, Moore's test). Between 1.5 and 3 m/s, normalized IAA sigmoidally decreased with airspeed significantly changing with each step (*$p<0.0001$, Moore's test, N=10; each colour represents one individual). This procedure constrained the normalized dataset between 1 and 0 for all bees, which enabled a ready comparison across different experiments over and above the inter-animal variability.

For freely flying bees, we estimated values of *groundspeed* vector (defined as the velocity of the bee relative to the fixed walls of the wind tunnel) as the rate of change of position of one of the antennal bases (point 2 from *Figure 1A*). Because the *windspeed* vector (velocity of airflow within the wind tunnel relative to the fixed walls) was under experimental control, we could calculate the *airspeed* vector (velocity of the bee relative to the surrounding air) as the vector difference between the *groundspeed* and the *windspeed* (*Figure 1B*).

In the experiments investigating the antennal response to temporal frequencies, we also analysed the IAA responses to calculate the threshold temporal frequency value for each bee after which it stops responding to any further changes in temporal frequency. We then binned the IAA values for each temporal frequency together to calculate a mean and standard deviation of the IAA response at each temporal frequency rate. We then compared these mean IAA to every other IAA value in the dataset for the bee using the paired non-parametric Moore's circular test in Oriana 4 (Kovach Computing Services, Anglesey, UK). The temporal frequency rate after which the mean IAA values are not significantly different from the rest of the IAA values is considered the threshold for that bee. The threshold for the bee shown in *Figure 2B* is 10 cps, for the bee in *Figure 2—figure supplement 1A* is 14 cps and for the bee shown in *Figure 2—figure supplement 1E* is 14 cps. The response to the first temporal frequency value is always distinguishable from the rest of the responses as the animal positions its antennae to changing optic flow (from zero to the first value). From the second optic flow onwards, the IAA changes as a response to the change in optic flow (from high to low or low to high) and not with the absolute value of the optic flow. Now that we know the threshold of the antennal response to optic flow, we can predict if for every change in temporal frequency value from one value to a randomly chosen value and so forth, the IAA should change significantly (*p<0.005, Moore's test, *Figure 2D*, *Figure 2—figure supplement 1C,G*; predictions shown in black, tick indicating that we predict that the IAA should change and cross indicating that the IAA should not change. The green dots indicate that the prediction is correct and the score is 1 in that case. The red dots indicate that the prediction is wrong and the score is 0 in that case. The collated scores are at the end of the series).We can also predict the direction of this change i.e. for a given change from one temporal frequency value to the other, should the IAA increase or decrease given that we know that the IAA should increase for every increase in temporal frequency values up to 10 cps of optic flow (*Figure 2D*, *Figure 2—figure supplement 1C,G1*; predictions shown in black, upward pointing arrow predicting that the IAA should increase and the downward pointing arrow predicting that the IAA should decrease. The green dots indicate that the prediction is correct and the score is 1 in that case. The red dots indicate that the prediction is wrong and the score is 0 in that case. The collated scores are at the end of the series). To test for the significance of this score, we scrambled the same dataset randomly so that there is no trend in the IAA values. We repeated the same process for this scrambled dataset. For all the individuals in *Figure 2A* and *Figure 2—figure supplement 1E and F*, our scores are significantly different from their scrambled datasets (*p<0.001; Student's T test) both in terms of the threshold and the direction of change in IAA. We repeated this process with a thousand scrambled dataset comparisons and the same results hold true in each case.

To understand how optic flow and airflow cues influence antennal positioning response in bees, we compared the mean interantennal (IAA) angles at various values of air speed and optic flow stimuli using the paired non-parametric Moore's circular test. In the experiments where we combined airflow and optic flow cues to look at antennal responses, we quantified these responses as the change in IAA (ΔIAA) from its value in Regime B, in response to step changes in speed of one or both stimulus. In these experiments, our ability to test various combinations of airflow and visual cues was restricted by two factors. First, each trial involving transition from Regime B to Regime A and Regime B to Regime C was performed within a single flight bout. Second, the on board memory of our cameras restricted our recording time to 68 s whereas each combination took a total of 60 s. This left us no more than three combinations that we could explore. In the *Figure 4*, we calculated the change in IAA (ΔIAA) as difference in the IAA from when only when one cue was being varied compared to when both the cues were present. In both these experiments, we compared all ΔIAA values with each other and with a hypothetical mean of zero using an ANOVA with a post-hoc Tukey's Honest Significant Difference test.

## Acknowledgements

The authors gratefully acknowledge Anand Krishnan, Tanvi Deora, Harshada Sant and Dinesh Natesan for their comments, and Umesh Mohan for technical advice, and our three referees for their thoughtful comments.

## Additional information

### Funding

| Funder | Grant reference number | Author |
|---|---|---|
| Air Force Office of Scientific Research | FA2386-11-1-4057 | Sanjay P Sane |
| Ramanujan Fellowship, Department of Science and Technology, India | SR/S2/RJN-40/2008 | Sanjay P Sane |

The funders had no role in study design, data collection and interpretation, or the decision to submit the work for publication.

### Author contributions

TRK, Planned and designed experiments, Conducted experiments, Analysed and interpreted data, Conception and design, Acquisition of data, Drafting or revising the article; SPS, Planned and designed experiments, Analysed and interpreted data, Conception and design, Drafting or revising the article

### Author ORCIDs

Sanjay P Sane, http://orcid.org/0000-0002-8274-1181

### Ethics

Animal experimentation: Study involved experiments on honeybees and were conducted according to ethical guidelines.

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
