## [Decision Letter]

Thank you for submitting your article "Multimodal speedometry in flying honeybees" for consideration by *eLife*. Your article has been reviewed by three peer reviewers, and the evaluation has been overseen by Ronald L. Calabrese as the Reviewing Editor and Eve Marder as the Senior Editor.

The following individuals involved in review of your submission have agreed to reveal their identity: Vivek Jayaraman, Emily Baird, and Jack Gray (peer reviewers).

The reviewers have discussed the reviews with one another and the Reviewing Editor has drafted this decision to help you prepare a revised submission.

Summary:

In this interesting study, the authors show that a linear combination of airspeed and optic flow influences antennal positioning in the bee. These results provide an instructive example of the integration of two distinct sensory modalities (mechanosensation and vision) for what is likely a reduction in perceptual ambiguity. Previous studies from the senior author and others have shown that antennae play an important role in flight control for many insects. Based on previous evidence, mechanosensation mediated by the Johnston's organ (JO) is important for sensing airspeed. There is also evidence (e.g., from Heran et al.) that JO sensing depends on antennal positioning, and, further, that this positioning depends on flight and wind speed. Altogether, these results make the study of multimodal cues that influence antennal positioning interesting. Rather than further clarifying the precise function of antennal positioning, the authors use inter-antennal angle (IAA) as an experimental tool to infer the insect's perception of its own speed. They then dissect the role of the two modalities in shaping the percept by monitoring antennal position while manipulating the bee's access to the distinct cues. The findings are in general well supported by the data and should be of interest to biologists who study multisensory integration and multimodal perception.

Essential revisions:

The reviewers were in agreement about the thoroughness and interest of the work, but request four revisions which must be addressed before this paper can be published in *eLife*.

1) Refocus the paper from speedometry to multimodal integration. Antennal position may well be a reasonable proxy for the bee's perception of speed, but the authors shouldn't claim the paper is directly about speedometry. The paper would be better framed as a study about multimodal integration, as assessed by antennal position.

2) Clarify that angular velocity is pertinent but was not analyzed. It is important to distinguish between angular velocity and temporal frequency and to make clear in the text that these were not studied independently. It is not necessary to do further experiments with different spatial frequencies.

3) Add more summary data with statistics to supplement data from individual bees.

4) The idea of cross-calibration of optic flow by airspeed was not convincing because of problems associated with head/tail winds.

These main points are amplified in the appended sections of the expert reviewer comments.

*Reviewer #1:*

Note that even if the experimenters could deduce increases and decreases in the bee's speed based purely on antennal position, which is very unlikely, these findings are not directly about speedometry – the insect does not itself use antennal position as a readout for speed, but may use it to keep the JO in its operating range (subsection “General relevance of antennal positioning response to mechanosensory and visual cues”, third paragraph). Thus, the strong emphasis on speedometry in the title, Abstract and Introduction is somewhat distracting and confusing.

Overall, the manuscript would benefit from being about a single, well-defined and properly motivated topic, and a well-defined set of questions with logical connections to the results. The main topic is perhaps the use of two different sensory modalities for speed perception as judged by a convenient (albeit imperfect) experimental proxy – antennal position. If that is true, it should be expressed more explicitly. The connection between speed measurements and IAA should then be explained in more detail in the Introduction or at the beginning of the Results section. Perhaps some of the information of antennal positioning presented in the Discussion (e.g. subsection “General relevance of antennal positioning response to mechanosensory and visual cues”, first and second paragraphs) could be moved to the Introduction or the Results.

*Reviewer #2:*

In general, the study approaches this topic in a very thorough manner, although I feel that there is one oversight that limits the scope of the findings. This main concern relates to the handling and interpretation of the results when the visual stimulus is presented. Previous work on honeybees has shown that they measure the angular rate of image motion independently from temporal frequency for visually-guided behaviour. The authors could have explored the respective roles of temporal frequency or angular velocity on the antennal positioning response by changing the visual stimulus from a grating pattern with a set spatial frequency to a pattern that contains a variety of frequencies (such patterns are not unusual and could have easily been used in the setup the authors used). As such, it is not clear in the data regarding the visual stimuli if the responses would be different if the angular velocity rather than the temporal frequency were varied. Although I do not think that it is necessary for the authors to do more experiments, I think that they should be more careful in their interpretation of the results and acknowledge that an analysis of the effect of angular velocity independent of temporal frequency would be informative as to the mechanism underlying the visual control of the antenna position.

Another concern that I have is that I do not understand the concluding argument that the bees use airspeed to calibrate optic flow. This is because airspeed may not provide very accurate information about the motion relative to the ground, such as in cases of tail wind or strong head winds, which honeybees do fly in. They either need to modify this statement or clarify how they predict the antennal positioning mechanism would work under these situations.

---

## [Author Response]

Essential revisions:

The reviewers were in agreement about the thoroughness and interest of the work, but request four revisions which must be addressed before this paper can be published in eLife.

1) Refocus the paper from speedometry to multimodal integration. Antennal position may well be a reasonable proxy for the bee's perception of speed, but the authors shouldn't claim the paper is directly about speedometry. The paper would be better framed as a study about multimodal integration, as assessed by antennal position.

We agree that our paper is not directly about speedometry, although the antennal positioning readout does provide specific insights into how their brain handles airspeed and ground speed data. However, as rightly pointed out by the referees, we are not showing its relationship to flight, but rather to antennal positioning. We take this point, and have altered the emphasis from speedometry to multimodal integration, as suggested. Specifically, we have changed the title, a key part of the Abstract, and shortened the last paragraphs in the Introduction that directly addressed speedometry.

2) Clarify that angular velocity is pertinent but was not analyzed. It is important to distinguish between angular velocity and temporal frequency and to make clear in the text that these were not studied independently. It is not necessary to do further experiments with different spatial frequencies.

We agree and have added a few sentences and a citation (Srinivasan, 1991) to explain this. Angular velocity is indeed the pertinent parameter, as shown by previous researchers. In our case, we have kept the spatial frequency constant, which means that angular velocity is the temporal frequency times a multiplication factor. We have added text in the Results section to add this caveat.

3) Add more summary data with statistics to supplement data from individual bees.

As required, we have added the summary figure (new Figure 2) which combines the behaviour data of all bees to linearly increasing and decreasing optic flow rates in the range of 0 to 25 cycles/s. We tried preparing a common figure for randomized data set, including the scores and reshuffled order of optic flow stimulus in each case, but this makes the figure very unwieldy, and adds very little to the overall point of that figure. Hence, we reverted to the single bee as an illustrative example, with all the other bees included in the supplementary information.

4) The idea of cross-calibration of optic flow by airspeed was not convincing because of problems associated with head/tail winds.

These main points are amplified in the appended sections of the expert reviewer comments.

Head/tail winds will alter the speed of the air-pocket in which the insect moves. However, our data show that what the insect detects as airspeed is its own movement within the air-pocket (i.e. relative to it). This is akin to a swimmer swimming within a current, where it can detect the relative flow of water immediately surrounding its body, but not the absolute speed. Thus, the ambient current (i.e. head and tail wind) should not affect this measurement, as indeed our data corroborates.

Reviewer #1:

Note that even if the experimenters could deduce increases and decreases in the bee's speed based purely on antennal position, which is very unlikely, these findings are not directly about speedometry – the insect does not itself use antennal position as a readout for speed, but may use it to keep the JO in its operating range (subsection “General relevance of antennal positioning response to mechanosensory and visual cues”, third paragraph). Thus, the strong emphasis on speedometry in the title, Abstract and Introduction is somewhat distracting and confusing.

We agree and have made the relevant changes (as also outlined in our response to the Point #1 of Essential Revisions above).

Overall, the manuscript would benefit from being about a single, well-defined and properly motivated topic, and a well-defined set of questions with logical connections to the results. The main topic is perhaps the use of two different sensory modalities for speed perception as judged by a convenient (albeit imperfect) experimental proxy – antennal position. If that is true, it should be expressed more explicitly. The connection between speed measurements and IAA should then be explained in more detail in the Introduction or at the beginning of the Results section. Perhaps some of the information of antennal positioning presented in the Discussion (e.g. subsection “General relevance of antennal positioning response to mechanosensory and visual cues”, first and second paragraphs) could be moved to the Introduction or the Results.

We agree and have moved part of this section to the Introduction, but kept the rest of the subsection “General relevance of antennal positioning response to mechanosensory and visual cues” in place. This change indeed makes the manuscript more streamlined and economically worded.

Reviewer #2:

In general, the study approaches this topic in a very thorough manner, although I feel that there is one oversight that limits the scope of the findings. This main concern relates to the handling and interpretation of the results when the visual stimulus is presented. Previous work on honeybees has shown that they measure the angular rate of image motion independently from temporal frequency for visually-guided behaviour. The authors could have explored the respective roles of temporal frequency or angular velocity on the antennal positioning response by changing the visual stimulus from a grating pattern with a set spatial frequency to a pattern that contains a variety of frequencies (such patterns are not unusual and could have easily been used in the setup the authors used). As such, it is not clear in the data regarding the visual stimuli if the responses would be different if the angular velocity rather than the temporal frequency were varied. Although I do not think that it is necessary for the authors to do more experiments, I think that they should be more careful in their interpretation of the results and acknowledge that an analysis of the effect of angular velocity independent of temporal frequency would be informative as to the mechanism underlying the visual control of the antenna position.

We agree with this comment, and have added a couple of sentences acknowledging that this is indeed the case, although keeping spatial frequency constant means that angular velocity is just a scaled version of temporal frequency, so the results should remain unchanged.

Another concern that I have is that I do not understand the concluding argument that the bees use airspeed to calibrate optic flow. This is because airspeed may not provide very accurate information about the motion relative to the ground, such as in cases of tail wind or strong head winds, which honeybees do fly in. They either need to modify this statement or clarify how they predict the antennal positioning mechanism would work under these situations.

A subtle but important point may be missed here. Airspeed refers to the speed *within* a given air pocket. When there are strong tail winds and head winds, the whole pocket moves, but the antenna only registers the movement within the air pocket. The net ground speed (optic flow) would be the vector sum of the airspeed of the bee within the air pocket and the speed of the pocket relative to the ground.